# A Modular AI-Driven Intrusion Detection System for Network Traffic Monitoring in Industry 4.0, Using Nvidia Morpheus and Generative Adversarial Networks

**DOI:** 10.3390/s25010130

**Published:** 2024-12-28

**Authors:** Beatrice-Nicoleta Chiriac, Florin-Daniel Anton, Anca-Daniela Ioniță, Bogdan-Valentin Vasilică

**Affiliations:** Department of Automation and Industrial Informatics, Faculty of Automatic Control and Computer Sciences, National University of Science and Technology Polithenica Bucharest, 313 Spl. Independenței, RO060042 Bucharest, Romania; anca.ionita@upb.ro (A.-D.I.); bogdan.vasilica@upb.ro (B.-V.V.)

**Keywords:** intrusion detection and protection system, event monitoring, networking, artificial intelligence, neural networks, Internet of Things (IoT), Industry 4.0

## Abstract

Every day, a considerable number of new cybersecurity attacks are reported, and the traditional methods of defense struggle to keep up with them. In the current context of the digital era, where industrial environments handle large data volumes, new cybersecurity solutions are required, and intrusion detection systems (IDSs) based on artificial intelligence (AI) algorithms are coming up with an answer to this critical issue. This paper presents an approach for implementing a generic model of a network-based intrusion detection system for Industry 4.0 by integrating the computational advantages of the Nvidia Morpheus open-source AI framework. The solution is modularly built with two pipelines for data analysis. The pipelines use a pre-trained XGBoost (eXtreme Gradient Boosting) model that achieved an accuracy score of up to 90%. The proposed IDS has a fast rate of analysis, managing more than 500,000 inputs in almost 10 s, due to the application of the federated learning methodology. The classification performance of the model was improved by integrating a generative adversarial network (GAN) that generates polymorphic network traffic packets.

## 1. Introduction

With a growing number of new devices connected to the Internet, it becomes increasingly difficult to develop and integrate a proper solution for supervising network infrastructure. Due to large data transfer volumes, threats are much more elaborate and difficult to identify in a timely manner. In this context, cybersecurity and data safety are even more important in specialized networks designed for entities manipulating critical information like the modern industrial sector, because vulnerabilities that are less obvious at first glance could lead to hazardous situations [1].

One class of technologies that can offer answers to the aforementioned issue is artificial intelligence. Papers in the scientific literature demonstrate that the efficiency of most cybersecurity solutions based on AI is influenced by the size of the input dataset and the models’ architecture that determines the output [2,3]. The more data the algorithm can work with, the more accurate the prediction will be. While the model itself can be better kept under control during the development process, the quality of the input is an obstacle that is harder to overcome and needs to be taken into consideration when AI is applied in a framework with a high level of criticality, such as Industry 4.0.

In Industry 4.0, the aim is to enhance performance and efficiency by integrating different technologies (Cloud, Digital Twin, Internet of Things, etc.) with industrial manufacturing equipment. As a result, the factory must be connected to services over the internet, and this can lead to important security problems [4]. With complex IoT infrastructures inside a factory, large volumes of data are exchanged between multiple working stations, and implicitly the human intervention is significantly reduced [5]. The diversity of components implies an equal variety of security vulnerabilities. Primarily, the computational failure rate could be increased by the AI elements used on an automated production line such as robots or cameras utilized for object recognition.

Although AI algorithms can have their own suite of vulnerabilities, they can also be the solution to this issue by monitoring network traffic within the industrial environment. Machine learning (ML) and artificial neural networks (ANN) have a wide area of application in this topic because they can be trained to predict or identify faulty situations [6]. One way to enhance security is to detect intrusion events using AI-driven monitoring events tools. In these categories systems are: intrusion detection systems (IDSs), intrusion prevention (IPS) system, and intrusion detection and prevention system (IDPS) are included. They utilize AI methodologies that make corelations between the fields included in the datagrams exchanged inside a network and the classes of behavior (e.g., normal behavior vs. abnormal behavior or different types of attacks) [7]. The fields labeled as dependent variables are usually the source and destination IPs, timestamp, protocols, flags, and timestamps.

Since the impact of unknown attack signatures on an IDS is essential, these kind of systems require constant updates, especially if they are part of an Industrial Internet of Things (IIoT) infrastructure. This is the main difference between a classical IDS solution and an AI-based IDS. Even if the AI algorithms are pattern-based, their classification capacities can be improved by using heterogeneous datasets for training and unsupervised learning methods. By creating derived polymorphic forms of the initial dataset, the topic of improving the system’s recognition capacity is covered and the neural network model responsible for the attacks’ classification is improved. An efficient mechanism to create polymorphic information is generative adversarial networks. First described by I.G. Goodfellow et al. [8], this artificial intelligence methodology is represented by two neural networks that train each other. Their training process is a competitive game based on an unsupervised deep learning method where the generator takes real data and adds noise to create similar inputs. Those inputs are analyzed by the discriminator which decides if they are real or not. Usually, GANs are used for generating media content, but their versatility also allows for the creation of polymorphic cybersecurity attacks by mutating the signature that existed in an initial dataset. This makes them suitable for being part of an AI-based IDS system as an additional tool for increasing the versatility of the classification mechanism.

On the matter of traffic classification techniques, the number of artificial intelligence algorithms that can represent an analyzer of an IDS is high. A detailed evaluation, made in Section 2, revealed that an IDS can use machine learning algorithms like vertical mode decomposition (VDM), support vector machine (SVM) or decision trees, as well as different classes of artificial neural networks. The algorithms are chosen based on the specification of the monitored system or by the expected results.

Taking all these into consideration, we identified three major points that an IDS for an IIoT infrastructure needs and, the goal of this article is to propose a solution that can cover all of them:Simplicity and flexibility of the process that integrates the IDS system into the industrial framework.Capability to constantly improve the IDS’s ability to recognize harm.The lowest possible score of false negative results.

Analyzing these approaches, the current study concluded that a suitable solution for the research’s purposes is an open-source software component skeleton that allows the implementation of cybersecurity attacks by utilizing neural network models. This framework is called Nvidia Morpheus and it covers all the points listed above. Arguments for that are supported by chapter 3 where an overview of the system was elaborated. Because Nvidia Morpheus integrates pre-trained AI models (like XGBoost) able to inspect the traffic across a network, it offers developers the possibility to integrate them into a solution by building simple pipelines, with models, preprocessing data approaches or inference procedures being encapsulated into modular blocks. The developer can also add their own modules. Therefore, to improve the model’s capability of classification, we proposed the integration of new modules, such as a generative adversarial network. Its purpose is to generate new traffic signatures by adding uniform noise through a PCAP (PCAP is an application programming interface destinated to network traffic packet capturing) capture.

### 1.1. Motivation

Because malfunction situations vary depending on the purpose of the Industry 4.0 environment, AI-based monitoring systems need to be trained on particular abnormal behaviors. Multiple studies try to evaluate the performance and protocols of intrusion detection systems in this new industrial revolution and to identify common aspects of an IDS for IoT ecosystems. Reviews such as [9,10] determine these common elements by analyzing a group of studies that refer to IDS based on AI for IoT. They focus on the comparison between supervised and unsupervised learning, the importance of the dataset for the classification processes, how different methods of detection can be applied, and what the impact of different common network attacks is.

Having such a broad array of Industry 4.0 frameworks, where new technologies are all included, together with a visible pending transition to Industry 5.0, which integrates new concepts of cognitive systems and human-machine interfaces [11], the importance of a network traffic monitoring component cannot be overstated. With such complex networks, the presence of a monitoring element, like an AI-based IDS capable of analyzing the internet traffic inside an IIoT infrastructure, is essential. This crucial role of cybersecurity and the investments in smart industries were the main motivations of this paper.

Therefore, it was important to investigate a variety of AI methods that are suited for the implementation of an IDS designated for the IoT industrial sector. Studies focused on intrusion detection methods conclude that only solutions specific to industrial systems’ needs are developed. These lead to niche-focused solutions that imply high costs.

Moreover, IDSs based on AI that are designated for modern industry may encounter bottlenecks due to the large amount of data that needs to be processed. These problems are related to both hardware and software components. To increase the efficiency of the identification process, the development of the system requires a suitable balance between software and hardware resource costs. We identified another important motivation: the lack of flexibility of intrusion detection systems and their dependence on the attack type.

With the goal of increasing their performance, most IDSs use a set of AI models that work together or are combined with different frameworks. In this context, metrics such as efficiency increase and computing cost reductions were tackled by the current paper. This is the reason why we decided to create a generic event monitoring solution for Industry 4.0 that can be modular and customized for specific IIoT environments.

Most AI-based IDSs that handle IoT data integrate machine learning algorithms, while neural networks and deep learning (DL) algorithms are not so popular due to their complexity [12]. According to studies, ML offers better accuracy results than other AI techniques (higher than 95%), even for real-time analyzis. However, they cannot integrate the flexibility of neural networks in improving their performance over time and their capability to efficiently manage new attacks. The lack of data can be another weak point, because researchers tend to train the neural networks on popular attacks like denial of service and evaluate the performance of AI algorithms on the detection of these kind of intrusions [13]. Furthermore, new technical and more configurable alternatives for building intrusion detection systems based on AI technologies are needed because they can be easily bypassed by inputs in which undetectable noise has been added. AI algorithms are still vulnerable to corrupted inputs, and some studies have demonstrated that imperceptible noise can radically change the output, an issue also confirmed by [8]. The possibility of realizing an AI-driven intrusion detection system that can be easily integrated into industrial infrastructures without adding a considerable quantity of resources became, in this context, an important objective of this research.

### 1.2. Contributions

This study presents the development of a generic model for abnormal behavior detection that can be used for network traffic supervision in different IoT infrastructures, especially Industry 4.0 platforms, just by changing the configuration of the integrated software modules. It aims to identify the possibility of implementing a configurable cybersecurity solution for event monitoring in an IoT network, being one of the first steps of a much larger ongoing research. Therefore, our contribution consists of the creation of two Nvidia Morpheus-based pipelines where all the important processes of an AI program, such as data preprocessing, inference and the correlation between the datagram’s features and the binary classification results are encapsulated into stages connected by edges.

Applying open-source AI models and frameworks that use graphics processing units for computation, we obtained an easily configurable AI-based IDS with a modular architecture. Part of the modules are integrated from the Nvidia Morpheus framework, while we added new modules for data conversion for PCAP capture files, data preprocessing, and data generation.

Another contribution is the increase in accuracy of attack detection by generating polymorphic attacks with the help of a generative adversarial network. The generated data are used for training and testing. The GAN module is created while upholding the modular architecture of the system, representing a new node. Finally, we created a preprocessing pipeline for this new heterogeneous input dataset and proposed an optimal IIoT platform that can integrate our IDS and make it able to be used at its full capacity.

In this manner, we significantly reduced development and resource consumption costs. Moreover, this will make the intrusion detection system more flexible and easier to integrate into industrial infrastructures.

### 1.3. Paper Structure

The current paper is structured in seven sections. Section 1 is the introduction, which contains generic information about the context of Industry 4.0 and the importance of intrusion detection systems in a smart industrial framework, together with the description of the desired AI-based solutions. Section 2 provides the related work and a brief literature survey, while Section 3 presents the theoretical background of this study, providing information about IDS/IDPS systems, Nvidia Morpheus, and the integrated AI algorithms.

The next three sections elaborate on the solution by describing in Section 4 the methods and algorithms used for implementation, in Section 5 the results of the experiments and in Section 6 discussions about the obtained results. The conclusions are presented in Section 7.

## 2. Related Work

Artificial intelligence mechanisms have been popular and present in almost every technological area in the last decade, even in the event monitoring domain. The literature proves that AI suits the intrusion detection field and that it can bring a number of advantages and solutions to long-lasting problems. Good results in intrusion detection operations are obtained by hybrid solutions. A hybrid solution incorporates multiple machine learning algorithms to implement a single IDS [14].

Klincer I.F. et al., 2024 [15] proposed a hybrid solution that implies the creation of a classification model based on vertical mode decomposition and statistics for processing the collected signal. An advantage of this study is that they succeeded in generating a dataset of attacks. Every signal collected from the network is transferred into the VMD, where it is decomposed into five coefficients. Representative features are extracted from these coefficients, which are afterwards filtered to eliminate unnecessary information. Only those features relevant to the discovery of an anomaly are kept. The output is then classified using different machine learning algorithms: k-nearest neighbors (k-NN), support vector machine (SVM), decision tree, and bagged tree (DT, BT). These algorithms obtained prediction scores higher than 90%, indicating a low level of false negative and false positive output.

Jasmin and Kurnaz, 2023 [16] presented a new method of deep learning by enforcing a convolutional neural network (CNN) in combination with biogeography-based optimization (BBO) algorithms and Sparse Auto Encoder Regulator. Their model is used for an IoT IDS implementation. The advantage of CNN is that this specific class of neural network combines the idea of learning algorithms with a network classifier [17]. In Jasim and Kurnaz, 2023 [16], the proposed architecture has multiple one-dimensional convolutional layers and Maxpooling layers for data training and prediction. With this deep learning methodology, they successfully obtained an accuracy higher than 99% with a training time of less than eight hours for their testing datasets. In the same direction, paper [1] proposed a solution for an IDS using another use case of feed-forward neural networks. They are using an extreme learning machine classifier. For better performance, the inputs go through a preprocessing operation, and only meaningful features are fed into the classifier. Similar methods of detection are proposed in [13], where a modified recurrent neural network creates gate recurrent units with the purpose of minimizing the prediction time.

All these intensely studied solutions have led to the development of specialized AI frameworks. The unsupervised learning mechanisms were of great interest. Their versatility lies in the fact that the neural networks they are based on can be capable of generating new inputs based on existing ones. Studies such as [18,19] describe solutions for IDS implementations built on GAN. These researchers are delving deeper into the problem, and they try to solve the computing limitations brought by ANN’s disadvantages by using federated learning (FL) as a training model for their GAN-based IDS proposal as in [20]. FL includes the ability of training an AI model across multiple nodes, such as different devices, servers, or virtual machines. Every node has the possibility to hold its own training information, so data is not transferred inside the network. Their model has a distributed architecture, so the tasks are processed on multiple machines due to the FL methodology.

Another interesting topic is real-time analysis and how this aspect is coming up with a lot of inconveniences. In Herrero et al., 2013 [20], the authors developed a hybrid IDS which was formed by an ANN, case-based reasoning, and a multi-agent system. They propose a multi-agent architecture where each component, part of the IDS, is described by another AI paradigm. The data capture and selection are very similar to the process described in this paper for the proposed IDS. The capture is filtered based on a set of fields and is afterwards converted into numerical information. The analysis process is done using Cooperative Maximum Likelihood Hebbian Learning and this aspect helps with the comparison operation.

Security Information and Event Management (SIEM), on the other hand, represents a solution for malware detection and event monitoring that centralizes security alerts coming from various sources. It manages all information in one place, has a reporting component, and logs the activity of the monitored system. SIEM is a system formed by several IDS/IPS components that work in a collaborative way.

Radoglou-Grammatikis, 2023 [21] describes a SIEM for an IIoT environment and how it is built on a software-defined networking paradigm (SDN), a method of centralization of the control logic in software-based and to separate this logic from hardware [2], honeypots, and AI algorithms. The proposed solution incorporates three IDPSs that use AI for detection, specifically a network-flow based IDPS, a host-based IDPS, and a visual-based IDPS.

A summary of this section can be found in Table 1 below.

## 3. Background

### 3.1. Intrusion Detection Systems

Over the years, various solutions have been designed with the aim of implementing a strategy to reduce costs and provide high-level security. In this category, we find intrusion detection systems (IDSs), intrusion protection systems (IPS), and their combination (IDPS). They can be passive or active; the difference is given by the way they react to an attack. The IDSs only notify when malware activity occurs, while IPSs and IDPSs try to defend against the attacks. During this period of continuous evolution, these systems have been facing problems regarding false negative results when a new attack occurs, lack of storage resources or issues with late response times. As such, a system using the old supervising and monitoring methods will not be able to offer full protection against modern malware. In the current context, for real-time analysis, where a significant quantity of information needs to be analyzed, this type of system has low intrusion detection performance. Starting from this hypothesis, the improvement in the intrusion detection field is a real need, and artificial intelligence (AI) is coming up with solutions.

IDSs and event monitoring systems are built to be capable of classifying a specific behavior as normal in terms of security policies or as a violation of them. In short, one of the main operating principles of an IDS is based on the binary classification principle. AI technologies like machine learning (ML) and deep learning (DL) offer solutions to the already mentioned issues. With AI algorithms, the accuracy of detection can be increased. Deep learning algorithms, and especially neural network models, have the powerful trait of being able to learn from previous outputs (in the case of an IDS, from previous attack detections) and adjust their results on the fly. Moreover, AI models can process large quantities of data, and, in fact, their level of performance is strongly associated with the size of the input set [22,23].

An IDS that does not return erroneous results currently does not exist, but cybersecurity researchers are trying to reduce the number of wrong outputs as much as possible. For example, IDSs based on artificial neural networks (ANNs) provide many false positives [24], while the oldest solutions are dealing with situations where they do not recognize a new attack.

The method of detection is another topic of classification for IDSs, and there are two categories as well: IDSs based on anomalies and IDSs based on attack signatures or patterns [1]. The signature-based IDSs store a set of attack patterns in a database. This database is verified, and the characteristics of the rules that define the attacks’ signatures are compared to the analyzed traffic capture. If a match occurs, then an attack is discovered. This kind of IDS is unable to detect those attacks that do not have a defined signature. A constant update of the signature database is needed for the signature-based IDS’s performance improvement; otherwise, the level of false negatives will grow [14]. On the other hand, anomaly detection techniques have a more generic approach and try to solve the problems regarding the inability to detect unknown attacks. Anomaly-based IDS builds a protocol that describes the normal behavior of the supervised system. In short, it will store descriptions of normal activities of the system as well as descriptions of harmful situations [1].

### 3.2. Nvidia Morpheus and AI Integrated Models

Nvidia Morpheus is an open-source AI platform that runs exclusively on the GPU (Graphical Processor Unit). Being a GPU-accelerated framework, Nvidia Morpheus can process large volumes of data in short timeframes, even real-time network traffic in cloud infrastructures. In comparison, data processing systems that run on CPUs cannot keep up with large volumes of information due to their architecture’s characteristics. CPUs can process parallel threads, but they are limited by the number of available cores. On the other hand, GPUs produce great results in parallel processing due to their increased core count and specialization of these cores. The scanning process, essential for an IDS, can be done in parallel, enabling a full network capture to be scanned at higher speeds [25].

Nvidia Morpheus has the capability to interface directly with the GPU and any application that integrates this framework inherits these advantages. This offers the possibility of developing a variety of AI cybersecurity solutions from passive applications to applications that prevent and manage threats. To improve the development process, Nvidia Morpheus uses a CI/CD (continuous integration/continuous delivery) model, which has a two-layer containerization system. Being an application destined for the cybersecurity field, when it starts, a command line interface is opened.

A new cybersecurity tool that integrates Nvidia Morpheus is built by creating software pipelines. The pipelines are directed acyclic graphs formed by configurable modules, named stages, which are interconnected through edges. Each stage organizes in nodes the operations they are responsible for, and they provide operations like data pre-processes, pre-trained AI models, ML algorithms, and post-processes output results. In this manner, a generic solution is more likely to be achieved due to the modular architecture.

Consequently, a Morpheus-based application can integrate a variety of AI models that are each stored in a specific container depending on their volume, while communicating with each other via virtual interfaces. Because of this, developers can integrate multiple technologies, algorithms, neural networks, and even other already existing AI frameworks by using the already created stages or by implementing new ones.

The AI models provided by Nvidia Morpheus are pre-trained and can be accessed by configuring the pipeline with the desired mode. The pipelines can be configured in two modes using one of the following models: Feature Importance Learning (FIL) and Natural Language Processing (NLP). NLP is a mode recommended for applications that are looking for text features. For example, it can be applied to identify keywords in an email for phishing attack recognition.

For IDS applications, FIL is more relevant because it uses federated learning. The federated learning technique manages data processing and training operations across multiple devices [20,26]. By this, Nvidia Morpheus arguments its capacity of processing even billion datagrams per second in a safety manner because the network traffic that needs to be analyzed is split between multiple secured devices. The FIL pipeline configuration mode will determine the Triton Interface Server to deploy the AI models and start the training procedure on multiple hosts. In the case of an IDS, AI models provided by Nvidia, such as Anomalous Behavior Profiling (ABP), are suitable. It can be used in a pipeline configured in FIL mode to detect abnormal features in traffic datagrams and predict the malware presence. The ABP model makes the classification by applying the XGBoost technique.

XGBoost is a gradient tree boosting method with great results in malware classification. The algorithm behind this method of classification is based on k additive functions for prediction, implying that the final model is a sum of smaller models. More precisely, the final model is a collection of simple decision trees, where each new tree is added sequentially to correct the errors of the previous prediction. Each tree is represented by a mathematical function, where the initial one is the mean value of the regression’s target. The new function computed in every iteration is the result obtained from the previous operation and the residual. At each step, the newly obtained model is trained to predict the residual [27].

XGBoost is a scalable machine learning system that runs faster than other similar methods due to its optimizations. This aspect is also linked to its characteristics because it reduces overfitting by utilizing parallel processing. Its initial algorithm proposal evolved over time and now integrates various decision trees [28], but an important element that remained is finding the best splits of the features [27]. This creates feature selection and reduces the data size by eliminating unnecessary elements. For a Morpheus pipeline whose purpose is data classification, it is important to configure the parameters that set the number of trees, the max depth, or the learning rate.

## 4. Methods and Materials

Cybersecurity solutions are developed based on the risk assessment of the system they are deployed to protect. An IDS developed for a personal computer will have different features and hardware components than an IDS developed for an embedded system or industrial networks. IDSs based on classical filtering algorithms are well known for their limitations in identifying unknown attacks, and while the ones based on ML and AI were able to overcome them, thanks to their prediction mechanisms, they have their own vulnerabilities. The current research investigated these issues, and the most common vulnerabilities have been detailed below. Consequently, an objective of the current research is to present an approach that can reduce the impact of these problems by extending open-source tools. The first step in this direction was represented by a classical hybrid IDS [29]. The initial study involved the creation of an IDS able to identify anomalies on a network as well as vulnerabilities on the host where it was installed. This system was developed using well-known algorithms and succeeded in identifying the important threats, but time efficiency results were unsatisfactory. Moreover, the signature database required periodic updates.

For the current presented solution, we applied Nvidia Morpheus because our objective was to test this framework and observe its capabilities in creating an IDS that can be integrated even in an Industry 4.0 infrastructure. The version used is the latest on the branch-25.02 from its GitHub repository (GitHub repository of Nvidia Morpheus framework: https://github.com/nv-morpheus/Morpheus, accessed on 1 May 2024).

The hardware infrastructure we used consists of two systems, each equipped with an Nvidia GeForce RTX 3050 GPU that can communicate with each other. To modify and run the application in our development environment, we were required to install additional software like Docker [30], Nvidia Triton Interface Server [31] with a version higher than 24.06, CUDA with at least version 12.1 (12.6 on the latest release), and the Nvidia Container Toolkit. A containerized workflow was chosen to avoid dependency issues when repeatedly modifying the source code, as the build scripts were always pulling the “latest” version of said dependencies from GitHub.

From a programming language point of view, the application is developed in Python 3.8 and C++ 17. These languages were chosen to maintain consistency with Nvidia Morpheus because our new modules were integrated into the framework. The newly developed modules were:

the conversion of the new PCAP captures into JSON format (JavaScript Object Notation). This format corresponds to the class of inputs that Nvidia Morpheus supports.the generative adversarial network encapsulated into a new node modulethe preprocessing for the polymorphic attacks.

For the new neural network models required by GAN, additional Python 3.8 library framework, such as Keras and TensorFlow kit were integrated. Therefore, the associated Python 3.8 libraries needed to be installed inside the Docker container used for development. These two libraries were useful tools in the implementation of the generator and the discriminator that create our GAN model.

### 4.1. Dataset Preparation and Preprocessing

The initial dataset has a total number of 1,134,528 records and is composed of two PCAP captures with 537,241 elements and 597,287 elements, respectively. The first PCAP capture is recommended by Nvidia Morpheus (Morpheus dataset stored in git-lfs: https://github.com/nv-morpheus/Morpheus/tree/branch-25.02/examples/data, accessed on 1 May 2024) to be adopted for the anomaly detection applications that integrate the XGBoost model. Being stored in a jsonline format, this capture dictated the inputs’ structure. The second set of data is represented by a part of an open-source network traffic capture done inside a SCADA (Supervisory Control and Data Acquisition) (infrastructure SCADA infrastructure dataset: SCADA/ICS PCAP files from 4SICS). Thus, a data sample was formed where the percentage for the benign behaviors vs. the abnormal is approximately 97% to 3%, as shown in Figure 1.

As the pipeline cannot use an input file in PCAP format, the data was processed, obtaining a final jsonline file format that included 13 attributes holding string data. The algorithm that converts this data (Algorithm 1) reads the PCAP binary file and generates a JSON object for each network packet by analyzing each layer of the datagram, its payload and header. The independent variables that describe a network packet are: timestamp, mac source address, mac destination address, protocol, source and destination IP (Internet Protocol) addresses, source and destination ports’ IP addresses, flags (e.g., Transmission Control Protocol (TCP) has flags that indicate the datagram’s purposes, such as “SYN” for synchronization packets or “ACK” for acknowledgment packets), length of the payload, and the payload, while the dependent variable, called “label”, has two possible values: “BENIGN” and “MALWARE”. Based on the independent characteristics, the classifier will identify the network flow as a normal or abnormal behavior by executing a binary classification.
**Algorithm 1: Converting the PCAP capture files into the input dataset****Data:   PCAP capture files****Results: JSON format file with the record characterized by 13 attributes**  Open the output file for writing**For** each file **f** in the PCAP file list **do**
Read the content of the file and store it in a network capture**For** each packet **p** in the network capture **do**
   1.Extract the independent variables (timestamp, flags, host IP, etc.)   2.Analyze the packet parameters   3.Label the packet   4.Create the JSON object   5.Write the JSON object into the output file

Return the JSON format file

Even though the presented IDS has a pre-trained AI model and data labeling is not mandatory in this context, it is required for the testing process. It is a difficult operation prone to errors, so it must be done by analyzing the datagram’s content as well as its header. In our case, we are looking at the protocol, the payload data, the sequence number, and the flags. For instance, if the protocol is ICMP (Internet Control Message Protocol) and the description of the packet contains “ICMP Activity”, this can be characterized as a ping flood or a scanning attack. If the protocol is TCP and in the flags list we identify synchronization without acknowledgment, this is categorized as a SYN flood attack. In conclusion, any kind of suspicious packet is labeled as “MALWARE”.

After the data collection is in the right format, the file that stores it is suitable as input for our first Nvidia Morpheus IDS pipeline. The preprocessing and data cleaning element are performed in a specialized stage. This stage cleans the data based on the configuration parameters of the pipeline and the record attributes’ values. Because fields of a JSON object are stored as strings, missing values are represented by the empty string or “null”. The difference is given by the data characteristics; for example, missing flags are replaced by “null”, while missing IP values are stored as empty strings. The conversion of the jsonline element is done by the already existing stage “FileSourceStage”. This stage parses the jsonline file into a data frame using the “cuDF library”. More technically speaking, the library transforms the string holding the JSON objects corresponding to the records into a structure capable of being manipulated on the GPU. A visual example can be observed in Figure 2.

This stage also starts the message flow of the pipeline. It encapsulates the new data structure into an Nvidia Morpheus message and transfers it to our pre-process stage. Inside the stage responsible for preprocessing, values stored in the data frame are mathematically converted into numerical data. The int value of the “flags” attribute contains a numeric representation that corresponds to a binary representation of the TCP telegram’s flags. Therefore, the int value is transformed into binary, and the column is encoded, resulting in five new columns with binary values inside them. Other operations are the timestamp conversion and detection of the packet flow based on the source and destination IP. At the end, the data corrections are performed. The null values are excluded and replaced with the mean value of the column. The stage responsible for the pre-process represents an important step in increasing accuracy because a quality input implies a low error rate. Consequently, the data used for training represents a critical component.

### 4.2. IDS Software Pipeline Architecture

A custom pipeline architecture consisting of two branches represents the software behind our proposed monitoring solution. The role of these pipelines is to connect different software elements for creating the processing and analysis components of the intrusion detection system. The main pipeline (Figure 3) has the role of analyzing the traffic captures and generating new attack signatures based on the analyzed ones, while the second one improves the performance of the system by testing the model with new generated inputs (Figure 4). Each pipeline is configured with a set of corresponding parameters that define the type of the pipeline, the processing rate (number of messages per second), the quantity of messages that can be handled, and which information from the message needs to be selected (number of independent fields). In the case of these two pipelines, the configuration is identical for both. The type of the pipeline is FIL because the goal of the IDS is the capability to process large volumes of data in short time frames. With this configuration, the distributed processing mode is applied. The rest of the parameters responsible for the pipeline’s performance were kept at the default values provided by Nvidia Morpheus (e.g., the pipeline batch size was kept at 100,000). In our application, already existing stages were reused, part of them were modified, and others were created from the ground up. For instance, the stage responsible for data preprocessing was adapted from an existing one in accordance with the input dataset (network traffic capture from an Industry 4.0 network), and the details were explained in the previous sub-section, while the stage responsible for attack generation was written from scratch. A challenge we faced in this phase of the implementation was the development of the input and output nodes of the stages. It is important to ensure compatibility between consecutive stages according to the type of messages that are exchanged. The number of output and input nodes can create errors that impact information processing during the pipeline’s progress. Each input information needs to be converted into a message type that is understood by the Nvidia Morpheus framework. In short, the output type of a stage must be supported by the input node of the next stage. To maintain this compatibility, it was important to keep the message payload consistent, even if it is modified and takes different forms across the pipeline.

The figures that represent the pipelines created for this solution (Figure 3 and Figure 4) use the following colour code: green marks the stages integrated directly from the Morpheus Nvidia framework, yellow marks the elements that were partially modified, and red marks the new stages created for our application.

For our application, the flow of information was created as the schema represented in Figure 3. The captured packets are stored in JSON files after parsing and are then passed through the input stage responsible for reading and converting the information into the message type known by Nvidia Morpheus that will travel through the pipeline until the last stages represented by classification and output.

After this first phase of data parsing, the values go through the deserialization stage, which applies part of the configuration parameters for the pipelines, and then it is transferred to the preprocessing stage where the data is transformed. The transformation process is done repeatedly while the information read from the jsonline file is sequentially fed into the pipeline.

After the data preprocessing operation, the information is passed through the newly implemented GAN stage. This stage contains a model represented by two neural networks (the generator G and the discriminator D), which are competitively training each other. The output of the network representing the generator is an input for the discriminator (see Figure 5).

Based on these aspects, the outputs of the model were stored in a new JSON file, and those records already stored in the input dataset were removed. The newly generated data has the same format as the process entities after all the conversion operations. This is an important aspect for the second pipeline of the application because it reduces unnecessary computation.

In this specific use case, the generator tries to create new attacks that are unknown to the discriminator. The goal of the system is to train the generator for minimizing the differences between original data and derived ones, and the discriminator for maximizing its capacity of labeling the outputs of the generator [8] as being real or fake. In this case, the generator used noise represented by randomly generated network traffic, which was additionally perturbated by adding simple mathematical operations. Random numbers from the interval [0,1], multiplied by a numerical anomaly factor value, represented the base in the construction of the noise. Both neural networks are described by linear models, and they use ReLU (Rectified Linear Units) functions and have just one hidden layer. The network weights used for training data were updated with the Adam optimizer, a more efficient implementation of the classical stochastic gradient descent procedure. The goal of using these generated attacks is to improve the performance of the deep learning system responsible for binary classification. A valid input is modified in an imperceptible way, but even the smallest perturbation can confuse these types of algorithms, and in the current case, this implies undetectable attacks.

After the new data is generated, the initial messages are passed through the inference stage, and the output of the inference passes through the classification stage. Unlike GAN, the models used for detection and interface are already developed, pre-trained, and integrated into the Morpheus framework. The inference stage integrates the Triton Inference Server, which provides large AI models; in our case, XGBoost. The interface stage starts a new thread, different from the one created by the pipeline, and splits the information into smaller “slices” that can be processed faster by the chosen pre-trained model. The inference stage passes output to the classification stage, which calculates probabilities (scores) that are transferred to the classification stage by indexing the labels. The role of this process is to determine the impact of each feature and how significant each of them is for the prediction operation. In the end, a probability that signifies what percentage of the analyzed record corresponds to a class is calculated. If this percentage is higher than the threshold for a specific class, then that input is assigned to that class. The classification stage by default has a threshold of 0.5.

More than just a threshold, the values of the configuration parameters play an important role in this step, and they should be chosen depending on the scenario where the IDS solution is applied. For the use case presented in this paper, where the analysis is done offline, the configuration did not have a major impact. However, in the case of online analysis, the situation is different. The most relevant parameters in the context of an online monitoring application are the pipeline batch size, the maximum number of batch size, and the number of internal threads. They influence the workflow of data processing on the GPU, targeting memory allocation and latency in data processing. Implicitly, the performance of the XGBoost algorithm is influenced by the process-management operation performed on the GPU. For example, the pipeline batch size determines the number of inputs that can simultaneously cross the stages, thus the number of samples that the AI model can handle during inference. 

At each step of the iterative XGBoost algorithm, the model is improved by adding a new tree that corrects the residual output. This new model is computed by also taking into consideration the learning rate, which is directly influenced by the performance of the GPU. If this number is not aligned with the model, the GPU’s memory is not properly allocated; therefore, operations like feature extraction are affected by latency. Moreover, the volume of input data will scale proportionally with the number of algorithm iterations, which can also lead to a lack of efficiency in GPU memory consumption. Essentially, the number of trees created during iteration is directly related to the number of samples that the model receives.

The pipeline also has an upper limit which determines the maximum number of messages/records that can be handled at the same time (the max batch size). It is important to have a proper balance between the max batch size and the pipeline batch size because if the value of the maximum batch size is too large compared to the value of the pipeline batch size, the GPU can run out of memory, while a value smaller than the batch size will cause multiple splits of the information during the inference stage. In the current case of offline analysis, we kept these parameters equal during the experiments, but in a real-life scenario, they need to be set based on the hardware configuration and the volume of samples.

After the results are computed, the information is recomposed into a human readable format, having an additional boolean label that specifies the type of the item as being a normal behavior or not. The serialization node reconstructs the information that is afterwards written in an output file using the writeToFile stage. Usually, the interface stage, deserialize, serialize, interface and writeToFile stages are indispensable in any Nvidia Morpheus pipeline.

The second pipeline has an experimental role in testing whether the capabilities of the IDS in data classification can be improved and how it handles derived forms of network traffic packets. Its main goal is to test the capability of the model to detect the addition of noise in an already known input. In this case, the pipeline flow is similar to the first pipeline, but it uses polymorphic inputs. The preprocessing stage works as a safety check step by cleaning the data of null or unknown values, and in this case, the classification stage is represented by an already implemented Morpheus stage.

As a conclusion to this chapter, Figure 6 graphically illustrates the entire data workflow and how our software solution works according to the details provided in the current chapter.

### 4.3. Monitored Hardware Architecture Prototype

While the software components reduce false negative outputs and the vulnerability of the proposed IDS to unknown data, the hardware elements provide support to increase performance and reduce resource costs. Because Nvidia Morpheus has the capability of interfacing directly with the GPU, the current IDS inherits these advantages. Therefore, both pipelines of our application generate processes that run in parallel inside the cores of an Nvidia GPU.

Even if the integration of Nvidia Morpheus can lead to an improved cybersecurity solution, the disadvantages are related to software portability. The IDS presented in this paper can only run on an Nvidia GPU, not on other types of graphics processor units.

While this IDS is made for monitoring an Industry 4.0 infrastructure, Figure 7 illustrates an example of a prototype environment that can be supervised with our Nvidia Morpheus-based IDS. This hardware infrastructure is composed of multiple work centers interconnected through an intranet network. A workstation can contain different components like industrial robots, cameras, sensors, or other processing units. It was considered that the best solution is to capture the data locally and afterward send it to a central processing unit, because usually, each workstation is responsible for a specific task. All data exchanged inside a work center is collected and centralized by a sniffer, which forwards it to a central unit for analysis. The work centers also exchange data, and this traffic can be collected in a similar manner. The central unit, where the entire traffic is collected, is represented by Nvidia GPUs and the current Nvidia Morpheus IDS. Depending on data quantity, this central unit can be adapted by configuring the IDS pipelines with more suitable parameters for the number of messages that need to be processed each second.

In the image below, we propose an infrastructure with two interconnected GPUs which communicate using a protocol developed by Nvidia called Nvidia NVLink. This kind of architecture is useful for critical systems or the ones which have a large amount of data that needs real-time processing. The first category requires two GPUs due to the redundancy principles and we consider this useful for the current proposal. If a GPU is down, the tasks can be forwarded from the broken GPU to the healthy one and it is usually used for those systems which are following the CIA paradigm (confidentiality, integrity, availability). Also, for both types, the number of bridges created between GPUs can vary from two to three slots. Inside the IDS the data will be divided between cores, and they will be analyzed in parallel as was previously described at the beginning of this section. In the end, the output is going through the final process and if an intrusion was discovered, the system will send a notification. At that moment the captured attacks are locally stored and analyzed offline.

## 5. Results

Based on the conducted tests, the current research proved that the proposed IDS offers a good level of protection against network attacks. These tests were performed by running the application on two systems with RTX3050 Nvidia GPUs. During tests, the focus was on the following aspects: the capability of the pre-trained models to identify real attacks, how efficiently the GAN model can be used for creating polymorphic attacks that can increase the IDS’s capabilities, the behavior of the IDS when analyzing polymorphic inputs and last but not least, how fast the input evaluation is done. As the adopted model was already trained to recognise a large set of sensitive and critical information, the current paper evaluated timeframe performance and how it can be improved as the model faces dynamic data provided by our GAN model [32].

A disadvantage of the Nvidia Morpheus framework is the visibility of the models. As opposed to other open-source AI solutions, the user does not have direct access to the neural networks’ architecture. Data splitting was not needed, due to the pre-trained model, hence this study used the entire data collection for testing because it aimed to verify if the model’s performance parameters are improved as the diversity of the inputs is increased.

The training of the generative model and the testing processes of pipeline 1’s (Figure 3) inputs are done in parallel. Therefore, tests revealed that the speed of the data classification in the “Nvidia Morpheus IDS_GAN pipeline” is drastically affected by the GAN training process, even if the solution is running on a GPU and the expectation was to obtain better performances related to their capabilities in the task scheduler. This can be observed by comparing the rate of processing in the Nvidia Morpheus IDS_GAN pipeline (pipeline 1) vs. Nvidia Morpheus polymorphic attacks pipeline (pipeline 2). Even if the preprocessing stage includes less mathematical operation, it is clear that the generation of new data inside the pipeline reduces the throughput of message processing. Table 2 presents the results and the comparison between a solution developed by Nvidia based on Morpheus (Nvidia Morpheus ABP pipeline) and the behavior of our two pipelines integrated in the current IDS. It was observed that the polymorphic inputs are rapidly analyzed, because the pipeline contains just Nvidia Morpheus original stages, and due to the input structure, it does not require preprocessing operations, like the transformation of some columns into dummy variables.

Another reason that contributed to the high score results (messages/second) of the second pipeline (Figure 4) is the system’s inefficiency in recognizing noise-affected data. The generated inputs are a polymorphic representation of the dataset, and the system labeled them as malware. During the experiments, considerable differences were not noticed. However, we observed that the polymorphic inputs can produce a high percentage of false negatives. Almost 99% of the generated inputs were labeled as anomalous network traffic packets.

The results of the IDS’s classification process were also influenced by the performances of the GAN model. When the rate of false negatives is high for the generated packets the accuracy and F1-score of GAN have unsatisfying results. While the discriminator can better classify the output from the generator, the pipeline responsible for analysing the data will do the same. The more epochs the GAN model will perform, the better the results will be. However, due to the chosen method, the stability of the training process is not perfect, therefore GAN produces a limited number of outputs. This is a weakness of the initial proposed GAN Framework, and it was solved in additional studies by adding different loss functions like Wasserstein distance [3,33] to the model.

Our classical GAN model has accuracy results between 0.21 and 0.87. One observation was that the accuracy is higher when the logarithmic losses of both generator and discriminator have similar values, and the number of epochs is higher than 1000. An epoch represents a training loop of the model or a round of the generator–discriminator game (see Table 3).

The overall accuracy of the IDS before starting to use the input-generated data was around 87%, after data were generated and the model integrated them, the performances were reevaluated as long as the GAN improved its capabilities. At each new step, the accuracy slowly increased which suggested that the model has a continuous learning mechanism. In the next figure (Figure 8a,b) are the results obtained after the IDS analyzed the first collection of generated data.

Having augmented data, the dataset becomes larger, more varied and in this manner, over time, the bias of the model can be eliminated. Even if the model is provided by the Morpheus framework, an improvement was noticed in the classification process. The classifier results are better if the GAN model increases its performance (see Figure 9). This was expected to happen, and we concluded that the models provided by Nvidia are using the inference server also for fine-tuning of the parameters of the chosen model. The difference between training data and our generated data contributes because it adds diversity to the dataset. The conclusion is that a good solution shall not only integrate the functions offered by the framework, but also their models.

## 6. Discussions

The development process reveals that Morpheus allows the integration of various AI models and different tools. Consequently, Nvidia Morpheus can be successfully integrated into various types of cybersecurity applications. Even though it was proven that it still requires configurations that should match the specific needs of each IIoT environment, these aspects do not imply the development of a new monitoring system from scratch. The monitoring system can be customized by building new stages. An excellent comparison that covers this topic is the one between the current IDS and its precursor. In the previous study, a hybrid IDS architecture was implemented [29]. The common point between the current study and the previous work is the adaptability of the system based on a set of configuration parameters, but the differences are more numerous. In the first study, the goal was to obtain a system that could offer protection on a network as well as on each host from the network. Moreover, the protected network did not contain IoT devices. The current study is targeted at the industrial sector; therefore, it requires better performance, an aspect that could not be covered by the IDS presented in [29]. The number of analyzed packets in [29] depends on the tasks that run on a host, while in the current paper, the analysis is independent of the monitored environment.

It was observed during the study that for a higher level of abstraction, the datasets need to be balanced and transformed into a format that can be understood as an input for the application. This leads to the main problem faced during the experimental and development phase and how these impact the efficiency of our solution compared to the ones proposed by Nvidia Morpheus in their examples. In the end the imbalanced data generated a high rate of false negatives and poor F-scores for GAN models, because the majority of the network packets used as inputs were labeled as normal behavior (Figure 1).

The main challenge was the integration of new software components into the framework and to create the compatibilities between nodes for being able to send the messages through the pipeline. Nvidia Morpheus is a framework recently released; therefore, projects that are focused on developing cybersecurity solutions using this tool are few. Moreover, the framework is unstable, and updates can produce incompatibilities inside the docker container or even in the software architecture. Due to this, studies done by an entity different from Nvidia are not documented in the literature yet. From this perspective it is hard to evaluate the capabilities of our solution in a larger context, but this represents part of our important contributions. From the specification point of view, the main disadvantage revealed by the results does not have an impact on the purpose of the system. In the case of an intrusion detection system, an error margin of false positives (even around 30%) can be acceptable, because suspicious behavior can be forwarded to an analyst. Furthermore, the system can be improved by extending the dataset both in the number of inputs and the variety of data. This represents the next step of our research, and further work can demonstrate these hypotheses.

On the other hand, the main advantage of this kind of cybersecurity solution is its versatility, although it is important to discuss its limitations. The experiments reveal that the large models offered by the framework for classification need to be customized and do not require considerable time for processing. For infrastructures that have special cybersecurity requirements, it can be applied because different data types can be analyzed if they are converted into message formats recognized by the framework’s module. The data inconsistency problem can be solved by two approaches, taking into consideration the Nvidia Morpheus’ characteristics discovered during the development process. The approach implies the implementation of specialized stages that can manage the data that is inconsistent. Moreover, the feature that implies the existence of a stage with multiple inputs can be used for injecting different data types into different flows.

## 7. Conclusions

During this research, an AI-based IDS solution was developed for monitoring network events inside a modern industrial infrastructure. By utilizing Nvidia Morpheus, the current solution involved the creation of two pipelines (shown in Figure 3 and Figure 4) for data processing. The XGBoost pre-trained AI models provided by Nvidia, together with the GAN model, improved the capabilities of the monitoring system to classify industrial network behaviors in a short time frame by analyzing over 1 million datagrams in 12 s. The theoretical study and tests, which implied the integration of the 4SICS datasets and its derived data collections generated with a GAN model formed by neural networks with one hidden layer, indicate that the framework is suitable for analyzing the network traffic in an industrial smart environment. The derived input dataset increased the accuracy of the XGBoost model by almost 3%, from 87% to 90%. In any test, this model did not obtain accuracy scores lower than 0.8. The derived data was processed faster as the time frame was approximately reduced by half, from ~39,466 messages per second to ~73,444 messages per second due to the elimination of the data encoder elements, which transformed fields like TCP flags into binary columns. The accuracy rate lower than 90% is influenced by three elements. The highest impact is given by the mathematical error rate, which is influenced by the manual data labeling process. The second aspect is the imbalanced data collection, where just 3% of the data represent abnormal network traffic. In the end, the message conversation done inside the GAN stage and the modified pre-procced stage affect the data quality, and implicitly, the XGBoost model’s performance.

Moreover, the study brings forth a proposal for a real use case of an industrial architecture where the IDS can be applied. The architecture presented in this study introduces an approach based on the theoretical background presented in Section 3.2 and the solution developed throughout the paper. The consumption of computational resources of the monitored system is reduced because our solution is running exclusively on the GPU and the capacity of monitored data could be increased in this way. The identification of unknown attacks was demonstrated to be improved by merging large pre-trained XGBoost with FIL configuration models with network packet flows generated by a GAN.

Due to the simplicity of the use case and the lack of diversity and volume of inputs, the model has a large rate of false negatives in the case of polymorphic inputs. The principal limitation that we faced was the capability of storing large volumes of data and having remote access to an industrial smart environment that could offer the opportunity to test the system in real-time. Taking into consideration the capabilities given by Morpheus, theoretically speaking, it is considered that the proposed IDS pipeline architecture can process even a live traffic capture (online monitoring), if the input file is replaced by a server that provides the traffic captures, such as a distributed event streaming platform like Kafka, different servers, or Cloud infrastructures. Therefore, the current IDS can be directly connected to live streaming networks, and it is configurable, being able to be applied on different hardware infrastructures just by changing the parameters for the max batch size and the pipeline batch size. This configuration is adapted to the requirements of each hardware architecture.

Even if our proposed IDS infrastructure has these theoretical abilities, in the current study, only offline analysis was tested. The reason for this is related to the current research phase of the project, and at this moment, the main purpose of this article is to create the core of our cybersecurity application using Nvidia Morpheus and test its classification performances.

From this point forward, we intend to improve the Nvidia Morpheus classification models by extending the dataset with more elaborate attacks and by eliminating the labeling rate error, because a pre-trained model should obtain better accuracy scores. Moreover, the research intends to follow a bunch of tests in a real smart industrial environment and test its performances on different configurations. The purpose of these tests is to identify their influence on the classification process in an experimental setting.

## Figures and Tables

**Figure 1 sensors-25-00130-f001:**
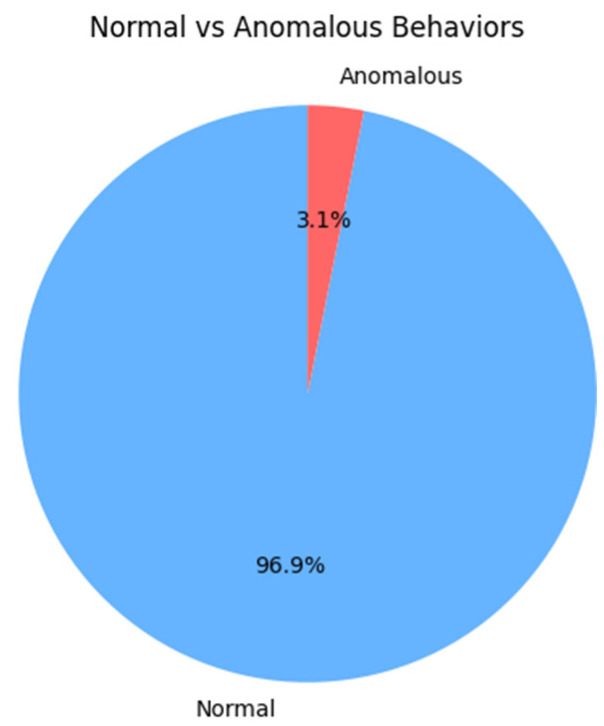
Distribution of normal vs. abnormal network traffic datagrams.

**Figure 2 sensors-25-00130-f002:**
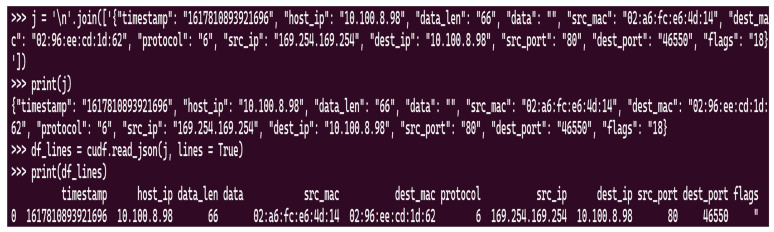
Example of a JSON object parsed into a data frame using cuDF.

**Figure 3 sensors-25-00130-f003:**
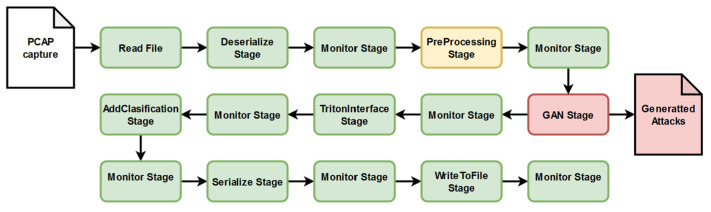
Main pipeline of the IDS created for classification of the PCAP capture and for generating polymorphic attacks.

**Figure 4 sensors-25-00130-f004:**
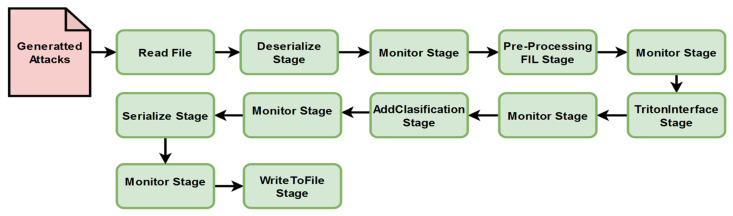
Pipeline for testing the performance of the classification stage with polymorphic inputs.

**Figure 5 sensors-25-00130-f005:**
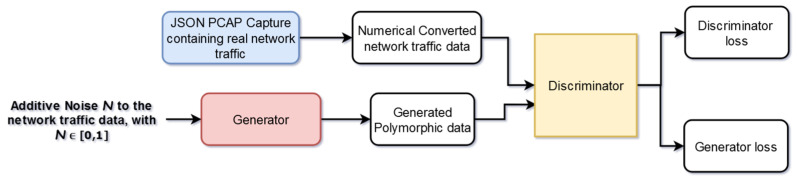
Generative adversarial network model implemented inside the stage specialized for generating polymorphic attacks.

**Figure 6 sensors-25-00130-f006:**
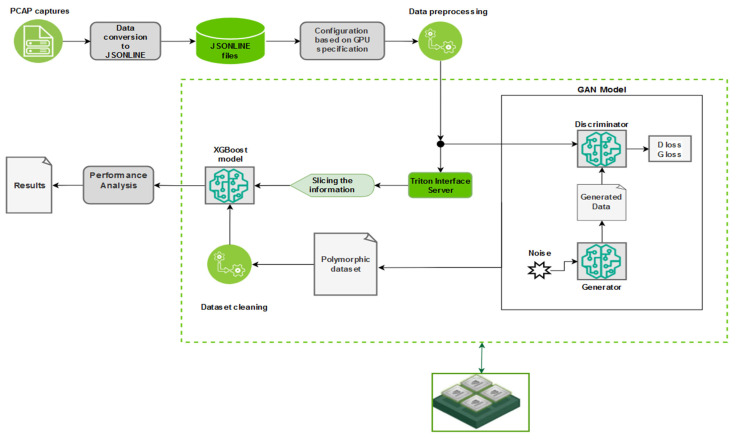
Workflow of the proposed IDS.

**Figure 7 sensors-25-00130-f007:**
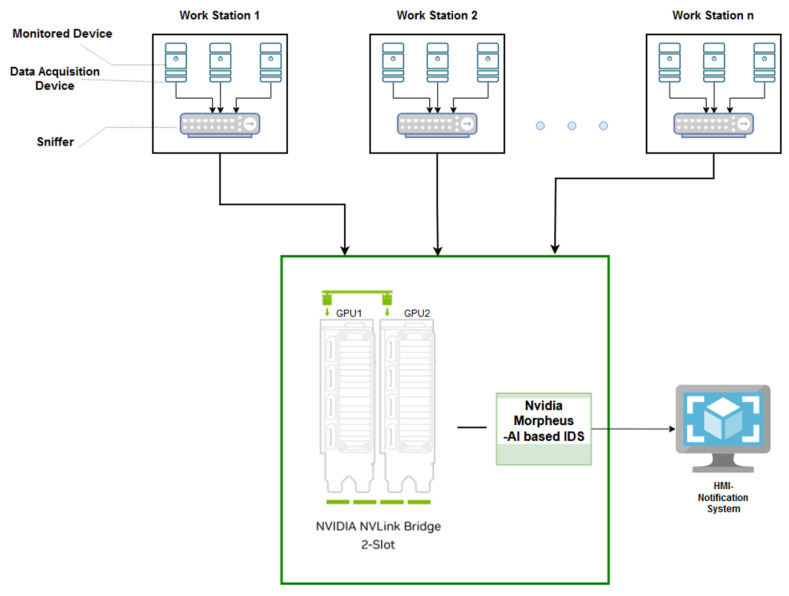
A prototype hardware architecture suitable for monitoring using an Nvidia Morpheus IDS solution.

**Figure 8 sensors-25-00130-f008:**
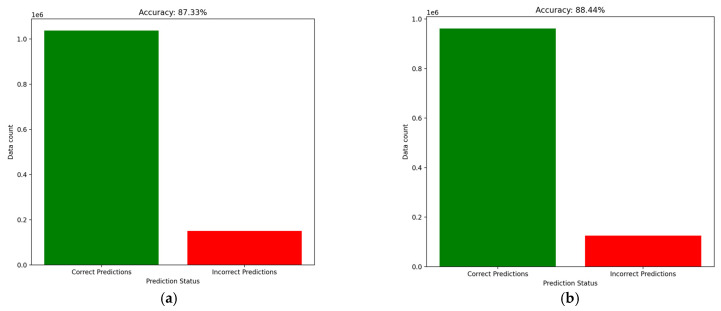
Models’ accuracy evolution before and after the first set of generated data. (**a**) Accuracy result before introducing generated data. (**b**) Accuracy results after generated data was introduced.

**Figure 9 sensors-25-00130-f009:**
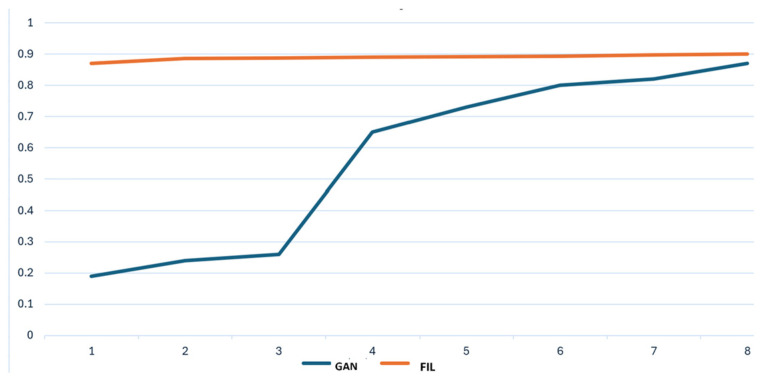
Models’ accuracy evolution.

**Table 1 sensors-25-00130-t001:** Comparison of the related work with the current article.

Research	Research Topic	Dataset	Methodology	Accuracy	Challenges and Contribution
Klincer I.F. et al. [15]	Switch port anomaly-based IDS	Original dataset created with data captured through switch ports. Dataset contains attacks with a high rate of frequency in Local Area Network (LAN).	Vertical mode decomposition applied to dataset; statistical feature extraction for data preprocessing. Hybrid method of classification using the following ML algorithms: SVM, k-NN, DT, and BT.	>90%	Combination of all ML algorithms for obtaining a classification framework. Original dataset. Lack of dataset diversity because the captures were made in the monitoring LAN.
Jasmin and Kurnaz [16]	Wireless Sensor Networks (IoT infrastructures)	CICIDS2017	Combining CNN with BBO algorithms and making an analysis of this hybrid methodology using different optimizers.	92–99%	Finding a hybrid solution with higher scores depending on the used activation functions and optimizers. BBO is slower than other classification methods.
Duy P. et al. [18]	Function prevention on adversarial samples of IoT attacks.	CICIDS2018 NSL-KDD	A framework that can create attacks using Wasserstein GAN and a black box IDS.	78–83%	Heterogeneous datasets. Generating polymorphic attacks.
Tabassum, A. et al. [19]	IDS for IoT networks focused on the targeted device attack.	NSL-KDD KDD-CUPP99 UNSW-NB15	A hybrid solution of federated learning (FL) and GAN networks used for classification.	92–99%	Heterogeneous datasets. Finding hybrid solution able to identify polymorphic attacks which reduce by proceeding the training process using in a distributed manner with the help of FL.
Herreo A. et al. [20]	A hybrid network-based IDS for attacks with impact on confidentiality, integrity, and availability (CIA model).	Real-time data	Combination of different AI paradigms for network traffic monitoring like case-based reasoning.	72–92%	Real-time monitoring defined by a time-bounded analyzer component.
Current Article	An IDS based on Nvidia Morpheus that integrates the XGBoost algorithms and GAN	4SICS Morpheus Network traffic examples	Combining Nvidia Morpheus XGBoost pre-trained model with the GAN framework.	87–90%	Improving the performances of pre-trained Nvidia Morpheus models for anomaly detection by generating polymorphic network traffic. Tested for offline analysis but capable of online monitoring due to Morpheus’ capabilities of processing large real-time data volumes.

**Table 2 sensors-25-00130-t002:** Comparison of three pipelines for malware detection from a timeframe performance perspective.

Solution	Time per Pipeline for Classification (Messages/s)	Accuracy	F1-Score
Nvidia Morpheus ABP pipeline	39,466.32	0.9014	0.0932
Nvidia Morpheus IDS_GAN pipeline	1240.72	0.8915	0.091
Nvidia Morpheus polymorphic attacks pipeline	73,444.72	0.3925	0.0665

**Table 3 sensors-25-00130-t003:** Evolution of GAN model’s results.

Evaluated Elements per Epoch	Precision	Recall	F1-Score	Accuracy	Epoch	D_Loss	G_Loss
Fake outputs	0.46	0.08	0.14	0.24	1000	0.71287	0.79833
Real outputs	0.20	0.70	0.32				
Macro avg	0.33	0.39	0.23				
Weighted avg	0.39	0.24	0.18				
Fake outputs	0.79	0.88	0.83	0.73	3000	0.73069	0.70071
Real outputs	0.45	0.29	0.35				
Macro avg	0.62	0.59	0.59				
Weighted avg	0.70	0.73	0.71				
Fake outputs	0.89	0.84	0.86	0.80	4000	0.70829	0.75026
Real outputs	0.58	0.68	0.63				
Macro avg	0.73	0.76	0.74				
Weighted avg	0.81	0.80	0.80				
Fake outputs	0.53	0.19	0.28	0.26	5000	0.68967	0.68131
Real outputs	0.17	0.48	0.25				
Macro avg	0.35	0.34	0.26				
Weighted avg	0.44	0.26	0.27				
Fake outputs	0.78	0.74	0.76	0.65	6000	0.76955	0.78994
Real outputs	0.33	0.38	0.35				
Macro avg	0.56	0.56	0.56				
Weighted avg	0.67	0.65	0.66				
Fake outputs	0.82	0.98	0.89	0.82	10,000	0.7978	0.7210
Real outputs	0.83	0.36	0.51				
Macro avg	0.83	0.67	0.70				
Weighted avg	0.82	0.82	0.80				

## Data Availability

The original contributions presented in the study are included in the paper; further details can be directed to the corresponding authors.

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
