# Peer review of "A Modular AI-Driven Intrusion Detection System for Network Traffic Monitoring in Industry 4.0, Using Nvidia Morpheus and Generative Adversarial Networks"

_sensors, 2024, doi:10.3390/s25010130_

Round 1
Reviewer 1 Report
Comments and Suggestions for Authors
1. Abstract: Revise the abstract to focus more on the primary issue being addressed and the proposed solution. Numerical results should be provided.
2. Introduction: Improve the introduction and try to make it more specific. Start with the motivation for the study, outline current challenges, introduce the proposed solution, highlight the main contributions in bullet points, and provide an overview of the paper's organization.
3. Literature Review (LR): In the Related Work section provide a table comparing existing solutions, highlighting the gaps your study addresses.
4. Method: The methodology section requires improvement:
· The current methodology follows a report (thesis) writing style; it needs restructuring to align more closely with a research paper format. More focus should be placed on the sections where your contributions will be.
· Provide proper justification for the choice of techniques, such as GAN. Explain why this technique is particularly suitable for your study.
· Clarify the dataset used: “For the current case, we utilised the dataset suggested by Nvidia for the testing process.” Specify the dataset name, provide details, and include a reference for the dataset.
5. Result and analysis: More analysis is needed to thoroughly describe the results. The current presentation primarily lists the obtained results without going into an explanation of why and how these results were achieved. Additionally, the poor F-score results require further explanation, particularly since, in IDS, accuracy alone is not the most critical metric. High accuracy can still result in a significant number of false positive errors, which must be addressed and analysed.
Comments on the Quality of English Language
Define all abbreviations upon first use to ensure clarity for readers. Revise the usage of abbreviations throughout the paper for clarity and consistency.
Author Response
Comment
„Abstract: Revise the abstract to focus more on the primary issue being addressed and the proposed solution. Numerical results should be provided.”
Response: The abstract was reformulated and it includes a summary of the motivation, details about the utilized algorithms and a short report about the results obtained during the testing phase (Lines 22-26: “The pipelines use a pre-trained XGboost model that proved to achieve an accuracy score up to 90%. The proposed IDS has a fast rate of analysis by managing more than 500000 inputs in almost 10 seconds, due to the application of the federated learning methodology. The classification performances of the model were improved by integrating a GAN model which generates polymorphic network traffic packets.”)
Comment
„Introduction: Improve the introduction and try to make it more specific. Start with the motivation for the study, outline current challenges, introduce the proposed solution, highlight the main contributions in bullet points, and provide an overview of the paper's organization.”
Response: The Introduction was significantly updated and it now presents the motivation in section 1.1. The Current context, challenges and solution are mostly described at the beginning of the introduction (Lines 32-107) sections. The solution is now described by adding new details (Lines 115-136). Then main contributions are mentioned in a dedicated sub-section (Lines 197-219) as well as the overview paper’s organization (Lines 222-232).
Comment
„Literature Review (LR): In the Related Work section provide a table comparing existing solutions, highlighting the gaps your study addresses.”
Response: Thank you very much for this suggestion. We introduced Table 1 that synthesizes several ideas from the scientific literature and facilitates the comparison based on research topic, data set, methodology, accuracy, challenges and contributions.
Comment
„The current methodology follows a report (thesis) writing style; it needs restructuring to align more closely with a research paper format. More focus should be placed on the sections where your contributions will be.”
Response:The paper was restructured having specific sub-sections like “Motivation”, “Contribution” and “Dataset preparation and pre-procesing”.
We also solved the requested issue by adding a special chapter named “Background” which contains all the theoretical information about the algorithms and the adopted methodologies. Part of this information was moved from “Methods and Materials”.
“Methods and material” suffered drastic changes in their structured and more details about how the development process was caried out were added in sections like “Data preparation and pre-processing”. The “Results” was extended with additional graphical test results and numerical information.
In “Conclusion”, we added numerical data and a short report for our methodologies and for our results.
Comment
„Provide proper justification for the choice of techniques, such as GAN. Explain why this technique is particularly suitable for your study.”
Response:The proper justification for the choice techniques is supported by the sub-section “Nvidia Morpheus and AI integrated models” which offers an introduction of the integrated technologies. Also, chapter “Material and Methods” is coming up with the advantages brought by each AI-model (e.g. Lines 515-519, Lines 583-596) and how they influenced our model. Moreover, the “Results”, “Discussion” and “Conclusion” chapters demonstrated the hypothesis made in the previous chapters and highlights once again the reason that are behind the decision of adopting methodologies like Nvidia Morpheus, GAN, XGBoost or FIL.
Comment
„Clarify the dataset used: “For the current case, we utilised the dataset suggested by Nvidia for the testing process.” Specify the dataset name, provide details, and include a reference for the dataset.”
Response:The dataset and how it was processed is now clarified in a dedicated part of chapter 4 “4.1. Dataset preparation and pre-processing”
Comment
„Result and analysis: More analysis is needed to thoroughly describe the results. The current presentation primarily lists the obtained results without going into an explanation of why and how these results were achieved. Additionally, the poor F-score results require further explanation, particularly since, in IDS, accuracy alone is not the most critical metric. High accuracy can still result in a significant number of false positive errors, which must be addressed and analysed.”
Response:The “Results” chapter includes more information about the testing process and additional graphical test results. Moreover, the reasons for our poor results were explained in section “Discussion” (e.g. Lines 755-761)
Comment
„Define all abbreviations upon first use to ensure clarity for readers. Revise the usage of abbreviations throughout the paper for clarity and consistency.”
Response:Thank you! We take this into consideration and the paper explains all the abbreviation when they first appear into text.

Reviewer 2 Report
Comments and Suggestions for Authors
- The title is Generic, and it has a Lack of focus and should reflect the core contributions and novelty of the paper while remaining concise and engaging. One suggestion can be: “A Modular AI-Driven Intrusion Detection System for Industry 4.0 Using Nvidia Morpheus and Generative Adversarial Networks”.
- The abstract should give a complete idea about the paper, so a brief of results or enhancements should be added.
- No clear novelty or contribution has been shown by the author, especially in (the introduction, methodology, and conclusions).
- The methodology section is unclear and should be clarified especially the used AI algorithms.
- Support the methodology by a flowchart of the work to illustrate the implemented system.
- Results should include a comparison with previous works to demonstrate the novelty or improvements of the approach.
- The conclusion should provide a concise summary of the findings, limitations, and future directions.
- Abbreviations should be defined the first time they are used (e.g., IDPS) and consistently used thereafter.
- The manuscript has some language weaknesses, so a copyeditor or proofreader is recommended to improve the flow and readability of the manuscript.
Author Response
Comment: „The title is Generic, and it has a Lack of focus and should reflect the core contributions and novelty of the paper while remaining concise and engaging. One suggestion can be: “A Modular AI-Driven Intrusion Detection System for Industry 4.0 Using Nvidia Morpheus and Generative Adversarial Networks”.”
Response: Thank you for your suggestion, we adopted the title proposed by you with a little modification. The new adopted title is: “A modular AI-driven Intrusion Detection System for Network Traffic Monitoring in Industry 4.0, using Nvidia Morpheus and Generative Adversarial Networks”
Comment: „The abstract should give a complete idea about the paper, so a brief of results or enhancements should be added.”
Response: The abstract was reformulated taking into consideration your suggestions and we included a brief of the results such as the value of the messages’ processing rate and information about the performances of the system. As well, the abstract mentions the methodologies that were utilized. For example: “The pipelines use a pre-trained XGboost model that proved to achieve an accuracy score up to 90%. The proposed IDS has a fast rate of analysis by managing more than 500000 inputs in almost 10 seconds, due to the application of the federated learning methodology. The classification performances of the model were improved by integrating a GAN model which generates polymorphic network traffic packets.”
Comment: „No clear novelty or contribution has been shown by the author, especially in (the introduction, methodology, and conclusions).”
Response: We explained the requested issue firstly by adding a paragraph named “Contributions” (Lines 195-219) in the Introduction. The paper’s structure was modified, by adding new sub-sections and by changing the order of the chapters. In the Motivation sub-section, we explained the reasons behind our study. Right now, the paper includes in “Introduction” a brief of the context, the motivation of the study, our contributions and a summary that explains how the paper is organized.
We also solved the requested issue by adding a special chapter named “Background” which contains all the theoretical information about the algorithms and the adopted methodologies. Part of this information was moved from “Methods and Materials”.
“Methods and material” suffered drastic changes in their structured and more details about how the development process was caried out were added in sections like “Data preparation and pre-processing”. The “Results” chapter includes more information about the testing process and additional graphical test results. Moreover, the reasons for our poor results were explained.
In “Conclusion”, we added numerical data and a short report for our methodologies and for our results.
Comment: „The methodology section is unclear and should be clarified especially the used AI algorithms.”
Response: As we mentioned in the previous comments, chapter 4 was re-written by adding more details about how the data collection were pre-processed and what datasets were used. We also described how the stages were modified and developed and we discuss about the algorithms that implements the utilized AI models. Moreover, the importance of the pipelines’ configuration parameters was explained.
Comment: „Support the methodology by a flowchart of the work to illustrate the implemented system.”
Response: We fixed this review comment by adding a new flow chart represented in Figure 6.
Comment: „Results should include a comparison with previous works to demonstrate the novelty or improvements of the approach.”
Response: We solved this comment in the beginning of chapter 6 “Discussions” (Lines 739-754) by offering information about the characteristics and results of our previous hybrid IDS.
Comment: „The conclusion should provide a concise summary of the findings, limitations, and future directions.”
Response: The “Conclucion” chapter summarizes the results offering numerical data and mentioning the challenges that we faced during the development process. The information about the used technologies were summarize. In the end this chapter brings up what are our further intentions by adding the improving topics that we would like to tackle in future (Fron Line 829 to the last line.)
Comment: „Abbreviations should be defined the first time they are used (e.g., IDPS) and consistently used thereafter.”
Response: Indeed, thank you, IDPS is now defined at its first use and other new added abbreviations have also been verified.

Reviewer 3 Report
Comments and Suggestions for Authors
The paper is well structured and clear from the content point of view. I find the subject of the paper in the contradiction with the summary section - as the subject is speaking about Industry 4.0 in general, which is quite generic and wide, and can be hardly researched within one paper, while the Summary section is giving more precise direction - that it is about IDS systems tackling in particular network traffic of Industry 4.0 environment.
Having said that, going forward, while Introduction part is "rich" with information about Industry 4.0 and its general infrastructure, it is not tackling the network part almost at all. Only some general info is given, and the focus has to be on the network infrastructure of the Industry, its logic and components.
The section related work is well structured and goes in enough level of details.
When it comes to the most important part: the sections of Method and Materials, as well as Results - this section is largely filled with "Related work"-like content. The concrete models developed by the authors are missing. The details description of the contribution is missing. The research methodology is not well described, as well as representative samples and detailed description of the use cases are missing. This section needs to be improved significantly.
Having in mind above, the concrete results in the Conclusion section need to be formulated, rather than wague statements. For example: "By utilizing Nvidia Morpheus, the 551 architecture of the solution implied the creation of two pipelines for data processing. The 552 pre-trained AI models provided by Nvidia together with the GAN model improved the 553 capacity of a monitoring system to classify network behaviors. The theoretical study and 554 tests made on the models of Morpheus indicate that the framework suits these problems 555 by obtaining a pipeline configuration that can process approximately double the number 556 of messages per second with an accuracy rate higher than 0.8." - which factors contributed mostly to the increase of the accuracy? What is the number of the learning iterations/cycles, and with which data complexity/patterns in order to achieve XY level of accuracy. This is exact science, where mathematical model can give 99% predictability on when you can expect that model will be trained if the quality of data is ABC...
The recommendation is to review all comments and adjust the paper before publishing.
Author Response
Comment: „I find the subject of the paper in the contradiction with the summary section - as the subject is speaking about Industry 4.0 in general, which is quite generic and wide, and can be hardly researched within one paper, while the Summary section is giving more precise direction - that it is about IDS systems tackling in particular network traffic of Industry 4.0 environment..”
Response: Thank you! The first change to be more focused was changing the title to “A Modular AI-driven Intrusion Detection System for Network Traffic Monitoring in Industry 4.0, using Nvidia Morpheus and Generative Adversarial Networks”. Then, the Introduction approaches the topic of network traffic in more detail and is now structured in sub-sectioned that explains the motivation of the study, the contribution and the paper’s structure. The entire paper has now a more structured and fluid workflow.
We added a special chapter named “Background” which contains all the theoretical information about the algorithms and the adopted methodologies. Part of this information was moved from “Methods and Materials”.
“Methods and material” suffered drastic changes in their structured and more details about how the development process was caried out were added in sections like “Data preparation and pre-processing”.
Comment: „Having said that, going forward, while Introduction part is "rich" with information about Industry 4.0 and its general infrastructure, it is not tackling the network part almost at all. Only some general info is given, and the focus has to be on the network infrastructure of the Industry, its logic and components.”
Response: As it was explained in the previous comment, the Introduction was significantly updated and it now tackles the network issues along with the motivation, challenges, and solution (Lines 108-232).
Comment: „When it comes to the most important part: the sections of Method and Materials, as well as Results - this section is largely filled with "Related work"-like content. The concrete models developed by the authors are missing. The details description of the contribution is missing. The research methodology is not well described, as well as representative samples and detailed description of the use cases are missing. This section needs to be improved significantly.”
Response: The related work content of this chapter was moved in chapter 3 named „Background”. The „Material and Methods” chapter has now more detailes about the algorithms, even pseudo-code representations. A new section that offers more detailes about the dataset is now part of this chapter and explaines how the dataset were processed. The overall functionality of the system is now explained in Figure 6. More detailes about the Ai algorithms and how they influence our system were also added.
The “Results” chapter includes more information about the testing process and additional graphical test results. Moreover, the reasons for our poor results were explained.
In “Conclusion”, we added numerical data and a short report for our methodologies and for our results.
Comment: „Having in mind above, the concrete results in the Conclusion section need to be formulated, rather than wague statements. For example: "By utilizing Nvidia Morpheus, the 551 architecture of the solution implied the creation of two pipelines for data processing. The 552 pre-trained AI models provided by Nvidia together with the GAN model improved the 553 capacity of a monitoring system to classify network behaviors. The theoretical study and 554 tests made on the models of Morpheus indicate that the framework suits these problems 555 by obtaining a pipeline configuration that can process approximately double the number 556 of messages per second with an accuracy rate higher than 0.8." - which factors contributed mostly to the increase of the accuracy? What is the number of the learning iterations/cycles, and with which data complexity/patterns in order to achieve XY level of accuracy. This is exact science, where mathematical model can give 99% predictability on when you can expect that model will be trained if the quality of data is ABC.”
Response: Thank you very much for this example, we followed it and Lines 790-809 demonstrate this aspect. Moreover during these lines, we explained the poor accuracy for our system even if it integrates pre-trained algorithms.

Round 2
Reviewer 2 Report
Comments and Suggestions for Authors
Here is the proofread version of your comments:
-
The Introduction section has improved and is good, but it is too long. Try to focus on the most important points.
-
Figure 6 (Workflow of the proposed IDS) in lines 615-616 needs more details.
-
The AI algorithm (XGBoost) used in the paper requires more details about it and the parameters that were applied.
-
In the Conclusion section, line 811, there is a mention of "(chapter 3)" which seems incorrect. Please check and correct it.
The paper is clear overall but needs minor improvements in grammar, sentence structure, and typographical errors to enhance readability and clarity.
Author Response
Comment 1: „ The Introduction section has improved and is good, but it is too long. Try to focus on the most important points.”
Response 1: Thank you for your suggestion, we reduced the dimension of the Introduction by deleting the redundant sentences and by rephrasing the information. In this moment the content of the chapter is formed by the information needed for creating the context of the paper.
Comment 2: „ Figure 6 (Workflow of the proposed IDS) in lines 615-616 needs more details.”
Response 2: Figure 6 was updated with a more detailed and graphical version.
Comment 3: „The AI algorithm (XGBoost) used in the paper requires more details about it and the parameters that were applied.”
Response 3: New theoretical data about XGBoost were added for creating the context Lines 397-404 and the influence of the pipeline parameters to the model was elaborated in chapter 4 Lines 664-692.
Comment 4: „In the Conclusion section, line 811, there is a mention of "(chapter 3)" which seems incorrect. Please check and correct it.”
Response 4: The sentence was rephrased and now it is clearer.
Comment 5: “The paper is clear overall but needs minor improvements in grammar, sentence structure, and typographical errors to enhance readability and clarity.”
Response 5: The paper was verified and grammatical errors were fixed.

Reviewer 3 Report
Comments and Suggestions for Authors
The authors took into consideration all remarks/comments, and significantly improved the quality of the paper. Now, it is sound scientific piece of research, with focused subject, very good methodology, results and conclusions derived.
Author Response
Thank you very much for all your comments and recommendations. We found your feedback very useful and we appreciated your implication during the review process. We hope that the revised version of the manuscript can be considered suitable for acceptance and we are readily available to provide further information as needed.
